# Targeted Therapeutic Options and Future Perspectives for HER2-Positive Breast Cancer

**DOI:** 10.3390/cancers14143305

**Published:** 2022-07-06

**Authors:** Angelica Ferrando-Díez, Eudald Felip, Anna Pous, Milana Bergamino Sirven, Mireia Margelí

**Affiliations:** 1Medical Oncology Department, Catalan Institute of Oncology-Badalona, Hospital Universitari Germans Trias i Pujol (HGTiP), 08916 Badalona, Spain; aferrandod@iconcologia.net (A.F.-D.); efelip@iconcologia.net (E.F.); annapousb@gmail.com (A.P.); 2Badalona Applied Research Group in Oncology (B-ARGO), Institut d’Investigació en Ciències de la Salut Germans Trias i Pujol (IGTP), Medical Departament, Universitat Autònoma de Barcelona, 08916 Badalona, Spain; 3AIDS Research Institute-IrsiCaixa, Institut d’Investigació en Ciències de la Salut Germans Trias i Pujol (IGTP), Hospital Universitari Germans Trias i Pujol, Universitat Autònoma de Barcelona, 08916 Badalona, Spain

**Keywords:** breast cancer, HER2-positive, estrogen receptor positive, triple-positive, HER2-targeted therapy, immunotherapy

## Abstract

**Simple Summary:**

The development of several antiHuman Epidermal Growth Factor Receptor 2 (HER2) treatments over the last few years has improved the landscape of HER2-positive breast cancer. Despite this, relapse is still the main issue in HER2-positive breast cancer. The reasons for therapeutic failure lie in the heterogeneity of the disease itself, as well as in the drug resistance mechanisms. In this review, we intended to understand the milestones that have had an impact on this disease up to their implementation in clinical practice. In addition, understanding the underlying molecular biology of HER2-positive disease is essential for the optimization and personalization of the different treatment options. For this reason, we focused on two relevant aspects, which are triple-positive disease and the role that modulation of the immune response might play in treatment and prognosis.

**Abstract:**

Despite the improvement achieved by the introduction of HER2-targeted therapy, up to 25% of early human epidermal growth factor receptor 2-positive (HER2+) breast cancer (BC) patients will relapse. Beyond trastuzumab, other agents approved for early HER2+ BC include the monoclonal antibody pertuzumab, the antibody-drug conjugate (ADC) trastuzumab-emtansine (T-DM1) and the reversible HER2 inhibitor lapatinib. New agents, such as trastuzumab-deruxtecan or tucatinib in combination with capecitabine and trastuzumab, have also shown a significant improvement in the metastatic setting. Other therapeutic strategies to overcome treatment resistance have been explored in HER2+ BC, mainly in HER2+ that also overexpress estrogen receptors (ER+). In ER+ HER2+ patients, target therapies such as phosphoinositide-3-kinase (PI3K) pathway inhibition or cyclin-dependent kinases 4/6 blocking may be effective in controlling downstream of HER2 and many of the cellular pathways associated with resistance to HER2-targeted therapies. Multiple trials have explored these strategies with some promising results, and probably, in the next years conclusive results will succeed. In addition, HER2+ BC is known to be more immunogenic than other BC subgroups, with high variability between tumors. Different immunotherapeutic agents such as HER-2 therapy plus checkpoint inhibitors, or new vaccines approaches have been investigated in this setting, with promising but controversial results obtained to date.

## 1. HER2-Targeted Therapy

### Introduction

About 15–20% of breast carcinomas overexpress the Human epidermal growth factor receptor 2 (HER2) [1,2,3]. HER2-positive (HER2+) breast cancer (BC) is a heterogeneous and aggressive disease. Despite the dramatic improvement after the introduction of HER2-targeted therapy [4,5,6], 15–25% of patients in the early stage will still relapse [4,5,7,8]. Several neoadjuvant studies reported that response to anti-HER2 treatment might be determined by endocrine receptor status and intrinsic molecular subtypes [9]. However, they do not fully recapitulate tumor heterogeneity and other biological features have been linked to response heterogeneity to HER2-targeted therapy and the risk of relapse. In recent years, research has focused on developing novel drugs, including anti-HER2 therapies with alternative and optimized mechanisms of action and others targeting new potential pathways to prevent and overcome resistance.

In the neoadjuvant setting, dual HER2-blockade has significantly improved the pathologic complete response (pCR) rate, a potential surrogate endpoint of improved survival, with ranges of 40–65%, depending on the treatment regimens and duration of neoadjuvant treatment [1,8]. Beyond trastuzumab, other agents approved for early HER2+ BC include the monoclonal antibody pertuzumab, the antibody-drug conjugate (ADC) trastuzumab-emtansine (T-DM1) and the reversible HER2 inhibitor lapatinib. Dual HER2-blockade administered in combination with chemotherapy has improved the survival of patients with metastatic BC. Beyond these, new agents have also shown increased progression-free survival (PFS) and overall survival (OS) in the metastatic setting, such as trastuzumab-deruxtecan (T-DXd) or tucatinib in combination with capecitabine and trastuzumab [10,11].

Beyond the HER2 receptor, other therapies to overcome resistance are being actively investigated in HER2+ BC. One approach has been blocking the cyclin-dependent kinases 4/6 (CDK4/6) as they are downstream many of the cellular pathways associated with resistance to HER2-targeted therapies with a key role in cell cycle and proliferation. Different trials have explored these strategies with encouraging and promising results, but definitive results are needed [12,13].

In addition, HER2+ BC is known to be more immunogenic than other BC subgroups, with high variability between tumors; thus, different immunotherapeutic agents are under investigation in this setting, with controversial results obtained to date [14,15].

This review reports the current and new therapeutic approaches being explored in HER2+ BC.

## 2. HER2-Positive Disease–Current Approaches

High expression of HER2 on cell surface of HER2 as been used as an ideal target by different mechanisms. Trastuzumab and pertuzumab are HER2-directed monoclonal antibodies (mAbs) inducing the recruitment of several immune cells with a subsequent activation of passive immunity, in combination with chemotherapy, these agents downregulate the oncogenic intracellular pathways led by HER2 activation via homo- and hetero-dimerization in the cancer cell membrane [16]. This happens because pertuzumab, by lying on the dimerization interface of HER2, prevents the formation of the the potent heterodimer HER2/HER3. This combined approach has shown synergy due to the relevant interactions between them and the immune system, leading to antibody-dependent cellular toxicity [17].

In recent years, HER2-directed ADCs have been developed using HER2 receptor to deliver cancer-killing agents inside the tumour cells with high sensitivity. For example, T-DM1 or T-DXd produce an increased cytotoxic effect in target cells and reduce target side effects [6]. These agents are molecules formed by an antibody linked to a chemotherapeutic agent, showing high efficacy due to the inherent activity of both the antibody and the chemotherapeutic agent but also, and especially in the case of new ADCs such as T-DXd, because they have an immune-modulator effect that affects neighboring cells [18,19].

Other options for targeting HER2+ BC include lapatinib, neratibin or tucatinib, small-molecule tyrosine kinase inhibitors (TKI). Lapatinib is an oral reversible inhibitor of the epidermal growth factor receptor (EGFR) and HER2 blocking the phosphorylation of tyrosine kinase residues. Consequently, it inhibits cell proliferation by regulating the mitogen-activated protein kinase (MAPK) and PIK3 pathways [20]. Neratinib, another oral irreversible inhibitor of EGFR, HER2 and HER4 [21], has been proven to induce cell cycle arrest and decrease HER2+ cell proliferation in preclinical studies [21]. Tucatinib is a selective and reversible HER2 inhibitor that potently inhibits signal transduction downstream of HER2 and HER3 via the MAPK and PI3K/protein kinase B (AKT) pathways [22]. Some studies have reported that patients who progress after trastuzumab might benefit from a TKI with or without trastuzumab, which may aid to overcome resistance [11,23,24,25,26].

Finally, a bidirectional crosstalk between HER2 and other receptors involved in BC such as Notch or TGF-β signaling, lead to resistance to different anti-HER2 treatments and involve higher aggressivity of HER2+BC disease. Although there are not current strategies under development targeting those pathways, their assessment might be useful as predictive and prognostic biomarkers.

### 2.1. HER2-Positive Early Breast Cancer—Current Approaches

Up to 25% of HER2+ EBC patients will relapse despite the existing strategies; there are still many opportunities for improvement in this setting.

#### 2.1.1. De-Escalation Approaches in the Early Setting

The pivotal HERA, BCIRG-006, NSABP B-31 and NCCTG N9831 trial results [4,5,27,28] established one year of trastuzumab (H) combined with three to six months of chemotherapy as the current standard adjuvant treatment for HER2+ EBC. The HERA trial showed no additional benefit after one year of targeted therapy [7]. Some strategies to avoid overtreatment of HER2+ patients have evaluated a shorter anti-HER2 adjuvant treatment. The Short-HER and SOLD trials evaluated nine weeks versus one year of H [29,30]. Non-inferiority of one-year treatment over the nine-week course was not demonstrated for disease-free survival (DFS) in any of these studies. The Short-HER trial showed favorable DFS (8.7 years of follow-up) in patients at low or intermediate risk of relapse in a *post hoc* analysis [31]. The PHARE trial also failed to demonstrate non-inferiority of 6 versus 12 months of H [32]. Finally, the phase III PERSEPHONE trial compared 6 versus 12 months of H and met its primary endpoint of non-inferiority in terms of DFS at four years. Nevertheless, OS results favored 12 months of H [33].

Some trials evaluated the de-escalation of chemotherapy strategies. Examples are the schedules without anthracyclines, such as the APT trial assessing the combination of 12 weeks of adjuvant paclitaxel with H for 12 months in patients with ≤3 cm, node-negative tumors [34]. The seven-year rates for invasive DFS (iDFS) of 93% and OS of 95% provided evidence to de-escalate chemotherapy in low-risk patients. The neoadjuvant scenario is a common strategy in HER2+ EBC. pCR correlates with patient outcomes and allows us to select subsequent therapies [35]. The TRAIN-2 study tested the use of anthracycline and non-anthracycline chemotherapy combined with pertuzumab (P) and H; no differences were obtained in pCR rates, three-year event-free survival (EFS) and OS [36,37]. The WSG-ADAPT trial evaluated dual HER2 blockade with P, with or without 12 weeks of paclitaxel. In the hormone receptor (HR) negative patients, the pCR rate was 90.5% with chemotherapy [38]. In addition, the KRISTINE trial compared T-DM1 plus P with Docetaxel (T), Carboplatin (Cn) plus HP, showing a lower pCR rate and higher risk of locoregional progression events with T-DM1 [39,40].

Furthermore, some trials explored the efficacy of chemotherapy-free combinations; TBCRC006/023, PER-ELISA and SOLTI-PAMELA achieved a 30% pCR rate with dual blockade with lapatinib (L) in combination with H but without chemotherapy [41,42]. The analysis of the intrinsic subtypes in the PAMELA trial provided pCR rates of 41% for the HER2-enriched (HER2-E) subtype. In contrast, in the NeoSphere trial, a 16.8% pCR rate was achieved in the arm of H plus P without chemotherapy arm [43]. The PHERGain trial explored whether omitting chemotherapy in PET (Positron Emission Tomography) responders was feasible. After two cycles of P and H, eight cycles were administered prior to surgery in patients with response according to PET and six cycles of TCHP in non-responders. Eighty percent of patients in the HP arm were responders and were spared chemotherapy, 40% of which achieved pCR [44].

#### 2.1.2. Escalation Approaches in HER2-Positive EBC

The intensification of therapy using dual anti-HER2 blockade has been another strategy in HER2+ EBC. The APHINITY trial randomized 4800 patients to receive adjuvant chemotherapy plus H with or without P for 12 months. Results showed a statistically significant but marginal benefit in terms of three-year iDFS, especially in node-positive patients, thus it is not routinely used [45]. Adjuvant dual blockade with chemotherapy in combination with L plus H was also tested, showing no improvement in outcome and presenting increased toxicity [46].

In the neoadjuvant setting, different intensification schemes have been tested. The KAITLIN trial evaluated the combination of T-DM1 plus P compared with taxane plus HP after anthracycline-based therapy in very high-risk patients. The results failed to find a reduction in the risk of invasive disease in the experimental arm [47]. The NeoSphere phase II trial evaluated dual blockade, randomizing patients to receive four neoadjuvant cycles of HT, PHT, PH or HT. In this trial, the dual blockade plus T than with HT, reported a higher pCR, with no added cardiotoxicity. Furthermore, despite the study was no powered to conclude the hypotehsis, PHT was associated with improved DFS, [43]. The TRYPHAENA and BERENICE trials analyzed dual HER2 blockade with HP and standard chemotherapy regimens with or without anthracyclines; they presented pCR rates of between 57 and 66% [48,49]. Different trials were also developed to evaluate dual blockade with HL plus chemotherapy in the neoadjuvant setting. In a recent meta-analysis including the GALGB 40601, Cher-LOB, NSABP-B41 and NeoALTTO trials, dual blockade was associated with an improvement in relapse-free survival (RFS) and OS, suggesting that the role of L should be reconsidered in EBC [50].

The functional assessment of treatment response in the neoadjuvant setting offers an opportunity to escalate therapy. Some trials evaluated the optimal treatment for patients who did not achieve pCR after neoadjuvant systemic therapy. In the KATHERINE trial, patients without pCR after neoadjuvant H and taxane-based chemotherapy were randomized to receive 14 cycles of T-DM1 or to complete one year of H. Treatment with T-DM1 was associated with improved three-year iDFS [51].

Other escalation strategies for higher-risk patients include the extension of adjuvant HER-directed therapies. As an example, the ExteNET trial, which randomized patients to receive one additional year of adjuvant neratinib or placebo after completing one year of H [52,53,54], showed an improved five-year iDFS in the neratinib arm. However, subgroup analysis found no benefit for HR-negative (HR-) patients and a statistically significant benefit in iDFS for HR-positive (HR+) patients. Treatment with neratinib was associated with diarrhea in 95% of patients and different strategies have been established to manage this side effect, improving the drug’s tolerability [55].

Table 1 and Table 2 summarize the main trials in the neoadjuvant and adjuvant settings.

### 2.2. HER2-Positive Advanced Breast Cancer—Current Approaches

Approximately one-third of HER+ patients are metastatic at diagnosis or will develop distant relapse. The introduction of HER2-directed therapies, such as monoclonal antibodies, TKIs, ADCs and, more recently, immunomodulation strategies, has improved the prognosis of patients with advanced breast cancer (ABC).

Nowadays, the recommended first-line treatment for patients with HER2+ ABC with “de novo” or relapsed disease one year after completing adjuvant therapy is based on the CLEOPATRA phase III trial. Patients were randomized to receive HT plus P or placebo [56]; after eight years of follow-up, there was an impressive increase in OS in the P arm. The ADC T-DM1 has been the standard second-line option in the last few years according to the results of the EMILIA trial [57], after demonstrating an improvement in both PFS and OS in comparison with L plus capecitabine [58]. The recently published DESTINY-Breast03 trial showed that treatment with T-DXd led to a impressive 72% reduction in the risk of progression compared with T-DM1 in patients previously treated with HT [59]. T-DXd is an ADC consisting of a humanized anti-HER2 monoclonal antibody connected via a tetrapeptide-based cleavable linker to a topoisomerase I inhibitor [60]. However, some toxicities were found, for example, TDXd was associated with interstitial lung disease and pneumonitis, but the incidence was lower than in previous trials with this drug. Drug-related interstitial lung disease was reported in 10.5% of patients receiving T-DXd; however, the number of grade 3 cases was low (2; 0.8%), and no grade 4/5 cases were reported. Nevertheless, careful monitoring is essential in its clinical use [61]. Based on these data, T-DXd will replace T-DM1 as the standard second-line therapy. Different trials have evaluated new strategies after progression with two lines of HER2-targeted therapy, but none have included T-DXd as second-line therapy.

Tucatinib, an oral TKI very selective for HER2 protein, was compared with placebo in combination in both arms with H and capecitabine in the phase III trial HER2CLIMB. This trial included a heavily pretreated population, patients with HER2+ ABC who had progressed after H, P and T-DM1, and included a significant proportion of patients with cerebral metastases. The HER2CLIMB results presented a significant improvement in PFS and OS [11,62]. Neratinib, an irreversible pan-HER TKI, was evaluated in patients previously treated with at least two lines of HER2-directed regimens, in combination with capecitabine compared with L plus capecitabine. The results showed a significant improvement in PFS but not in OS [23]. The SOPHIA trial, another phase III trial that included patients with progression after at least two lines of anti-HER2 therapies, randomized patients to receive chemotherapy plus margetuximab versus chemotherapy plus H. Chemotherapy plus margetuximab showed a significant improvement in PFS but not OS [63]. The ADC (vic)-trastuzumab duocarmazine was also compared with physician’s treatment choice in patients with at least two prior lines of therapy or previous T-DM1 treatment. The results showed a significant improvement in PFS but not OS. Treatment with (vic)-trastuzumab duocarmazine was associated with 78.1% of eye toxicity, 21.2% of cases being grade ≥3 [64]. The combination of L plus H showed an improvement in PFS and OS in comparison with L in heavily pretreated patients before the incorporation of P and T-DM1 into standard clinical practice [65].

Considering all these results, T-DM1 or tucatinib-based combinations seem to be the most interesting alternatives for HER2+ patients. Neratinib, margetuximab, L or H in combination with chemotherapy might be considered as later-line options.

### 2.3. HER2 Heterogeneity as an Opportunity to Select the Best Therapy

The heterogeneity of HER2 BC provides clinicians with a unique opportunity for therapy individualization and personalization.

For example, several neoadjuvant trials and meta-analyses in HER2+ disease have demonstrated a solid association between HR status and pCR, reporting better outcomes in HR- tumors [35,66,67]. Beyond HR status, intrinsic molecular subtypes defined by PAM50 [68] can be identified in HER2+ tumors, being HER2-E the most frequent (~47%), followed by Luminal B (~18–28%), Luminal A (11–23%) and Basal-like (7–14%). Additionally, the impact of HR status in this distribution shows the HER2-E subtype representing 75% of the HR- HER2+ tumors but only 30% within HR+ HER2+ [69,70]. Different trials, a systematic review and a meta-analysis have confirmed the association between the HER2-E intrinsic subtype and a higher pCR [66]. In the adjuvant setting, the Short-HER study’s molecular analysis showed that the HER2-E subtype was associated with worse metastasis-free survival in both therapy arms [71]. However, the biological heterogeneity of HER2+ tumors might also be related to the immune response after treatment induction of microenvironmental changes. The prognostic and predictive role of tumor-infiltrating lymphocytes (TILs) has been evaluated in clinical trials involving HER2+ patients [23,62,63,64,72,73,74]. The integration of both clinical and biological heterogeneity is an interesting concept for deciding HER2+ disease treatment. In this sense, tools such as HER2DX have been developed in EBC, which include tumor size, nodal status, stromal TILs, PAM50 subtypes and the expression of 13 genes [75]. Using genomic and clinicopathological data from patients included in the Short-HER trial, this signature has been validated as a continuous variable in independent datasets from adjuvant and neoadjuvant trials. Interestingly, this signature has shown significant association with DFS and distant metastasis-free survival (DMFS), suggesting that treatment could be de-escalated in some HER2+ EBC patients.

The integration of molecular data for each patient might lead to therapy personalization and avoid unnecessary toxicities, as well as toward the development of more cost-effective treatments.

## 3. The Role of the Estrogen Receptor in HER2-Positive Tumors

Despite the obvious biological difference, HER2+/ER+ tumors are not, yet, treated differently from HER2+/ER- BC, and endocrine therapy is only added as maintenance after a standard anti-HER2 agent in combination with chemotherapy. However, in the last years, the effort has been focused on finding combinations to improve chemotherapy regimen toxicity. One example is the PHERgain phase II trial, in which 67% of the patients were ER+. A trastuzumab plus pertuzumab chemotherapy-free regimen was evaluated, and hormone therapy was added in ER+ patients. Although the chemotherapy-free regimen had a significantly lower pCR rate after the neoadjuvant regimen, 35.4% of the patients achieved a pCR after neoadjuvant treatment. Interestingly, with the addition of endocrine therapy, ER+ status was not a predictor of treatment response. However, the HER2 immunochemistry score had an impact on pCR: a HER2 3+ immunohistochemistry score achieved a higher pCR rate than tumors with a 2+ score [44]. Another example is the PERTAIN phase II trial, that combined an aromatase inhibitor (AI) (anastrozole or letrozole) plus trastuzumab, with or without pertuzumab in HER2+/ER+ metastatic BC patients with no prior systemic therapy. The AI plus trastuzumab and pertuzumab group had a significant increase in mPFS, 18.89 vs. 15.80 months (hazard ratio, 0.65; 95% CI, 0.48 to 0.89; *p* = 0.007) [76]. Understanding the biological crosslink between the HER2 and ER pathways will help improve treatment and, therefore, patient outcomes in the future. 

### 3.1. Crosslink between HER2 and Estrogen Receptors

ER and HER pathways are known to be frequently deregulated in BC [77]. When ER is not expressed, the HER2 pathway is associated with a more proliferative BC and an immune activation stroma with elevated TILs. Deregulation can occur by upstream signaling molecule alteration or due to a genetic/epigenetic change in downstream signaling molecules. ER acts as a hormone nuclear transcription factor and binds specific DNA sequences to promote the proliferation and survival of genes [78]. Furthermore, ER is involved in non-nuclear pathways and can interact at a cytoplasmic level with several tyrosine kinase receptors (e.g., HER2) and downstream signaling factors [79]. These two ER pathways interact at several levels with many cellular kinase networks to undergo bidirectional crosstalk that increases the ER signaling and kinase-related pathways [80]. Additionally, ER can interact with HER signaling members via G protein interaction [79] (Figure 1).

An in vitro study using BC cell lines described significantly lower ER levels when the HER2 gene was transfected, presenting an inverse correlation between the levels of HER2 overexpression and ER [81]. Moreover, it has been suggested that the HER signaling factor could reduce both mRNA and protein levels of ER expression. BC subtypes based on the PAM50 classification within HER2 tumors differ according to their ER status. Interestingly, more than two out of three HER2+/ER+ tumors are luminal A or B, when analyzed via the PAM50 signature. These are ER-dependent tumors with low activation of the HER2/EGFR-pathway [82]; this subtype benefits less from anti-HER2 treatments [83]. The survival prognosis and time to relapse after radical treatment differs between HER2/ER+ and HER2+/ER-. For example, overall survival five years after EBC diagnosis was significantly higher in HER2+/ER+ (88.2%) compared with the HER2+/ER- subgroup (83.9%) [84]. After five years, the recurrence risk in the HER2+/ER- group decreased progressively, however it was conserved over time in the HER2/ER+ group [7]. Based on these findings, in the future, ER+/HER2+ BC might be defined as a distinct molecular disease [85].

Beyond ER itself, an increasing number of studies have also suggested that the crosstalk between prolactin receptor (PRLR) with ER and EGFR/HER2 pathways plays a crucial role inducing tumour proliferation and thus be associated with resistance to anti-HER2 treatment and with higher risk of relapse [86]. For this reason, combined treatments targeting PRLR and endocrine therapy with HER2 targeted treatments have been suggested as promising strategies to overcome potential mechanisms of resistance [87].

### 3.2. Efficacy of Anti-HER2 and Endocrine Therapy in ER+/HER2+

Due to the crosstalk between the ER and HER2 signaling pathways, therapeutic strategies blocking only one pathway result in the upregulation of the other, leading to treatment resistance [88,89]. HER2 overexpression has been related to de novo and acquired resistance to endocrine therapies [89]. Multiple trials have demonstrated the lower pCR rates in HER2+/ER+ compared with HER2+/ER- patients using chemotherapy and dual anti-HER2 target therapy in the neoadjuvant setting [9,39,90]. 

A possible strategy to improve treatment performance in HER2+/ER+ would be to prolong anti-HER2 treatment. As an example, the TBCRC023 phase II trial, in which 65% of patients were HER2+/ER+, reported that increasing the length of anti-HER2 treatment (trastuzumab/lapatinib) from 12 to 24 weeks increased the pCR rate (9% vs. 33%) in HER2+/ER+ patients who also received Letrozole. Surprisingly, this benefit was not seen in HER2+/ER- patients [41]. In the same direction, the ExteNET trial showed that adding sequential anti-HER2 treatment with trastuzumab and Neratinib for two years was beneficial in HER2+/ER+ but not in HER2+/ER- patients [52]. An explanation of this fact could be that the continued blocking of both HER2 and ER pathways would induce a decrease in proliferation and apoptosis. In this sense, an in vitro study demonstrated that continuous dual inhibition of the HER2 pathway with Trastuzumab plus Lapatinib led to ER function as a critical escape/survival pathway in HER2+/ER+ cells, thus, a complete blockade of HER2 together with the ER pathway may be required for these patients [91]. 

Multiple efforts have been made to improve response rates, time to relapses and survival, including improved cancer patient selection. For example, when ESR1 and ERBB2 expression were quantified in the NeoALTTO trial, high levels of ERBB2/HER2 and low levels of ESR1 were associated with higher pCR rates in all treatment arms [92]. Also, in the adjuvant setting, NSABP-31 trial analysis succeeded in clustering patients into three groups using an eight-gene signature regarding ESR1 and ERBB2 expression. As expected, the low-intermediate ESR1 and high ERBB2 groups received the most significant benefit from adjuvant treatment using anti-HER2 therapies [93]. Regarding intrinsic subtypes based on PAM50, many studies demonstrated that an HER-2 enriched (HER2-E) subtype predicts pCR after HER2 neoadjuvant treatment [94]. Surprisingly, ER status by immunochemistry loses the association with pCR when the PAM50 subtype is considered; therefore, intrinsic subtypes may better reflect sensitivity to treatment than BC cell biology. In HER2+ and HER2-E tumor cell lines, treatment with anti-HER2 drugs induces a low-proliferative luminal phenotype. Interestingly, suspension of HER2 therapy in vitro reverts them to the original HER2-E phenotype, suggesting that HER2-targeted therapy should not be stopped despite resistance. These changes were more evident in ER+ disease, probably due to ER and HER2 signaling crosslinking [95]. Another signature used in HER2+/ER+ tumors is Rbsig, which analyzes 87 genes to build a signature related to RB1 loss-of-function. Higher Rbsig scores were found to correlate with a higher response to chemotherapy whereas low Rbsig expression identified HER2+/ER+ patients with low pCR rates. These findings suggest that patients with a low Rbsig may be potential candidates for a de-escalated regimen with HER2-targeted therapy plus hormonal therapy and CDK 4/6 inhibitors [96]. Finally, high Rbsig HER2+/ER+ tumors have a lower tumor-infiltrating lymphocyte (TIL) rate than high Rbsig HER2+/ER-. However, a higher TIL rate is associated with increased pCR rates after neoadjuvant treatment in both HER2+/ER+ and HER2+/ER- tumors [97].

### 3.3. Future Combination/Ongoing Studies to Improve the Efficacy of Endocrine Anti-HER2 in ER+/HER2+

Understanding the molecular biology and resistance mechanisms in HER2 and ER BC is necessary for the optimization of future drug combinations in HER2+/ER+ disease. For example, one-third of HER2-/ER+ metastatic patients carry a PI3KCA mutation [98,99]. The role of hyperactivation of the downstream PI3K-AKT pathway has been broadly studied and associated with resistance to HER2-targeted therapy in preclinical models, especially trastuzumab [100,101]. PI3K-AKT pathway activation might be induced by PTEN loss-of-function or a PI3KCA mutation, which are more frequent in HER2+/ER+ patients than in HER2+/ER-. [82]. After progression to anti-HER2 treatment, some trials have attempted to block PI3K-AKT using mTOR or PI3K inhibitors in combination with anti-HER2 treatment. Alpelisib plus TDM-1 in heavily pretreated HER2+ patients has shown promising results with an overall response rate of 43% [102]. Nevertheless, the combination of PI3K inhibitors with anti-HER2 drugs seems to induce severe side effects and most trials are still in the early phases, aiming to establish the maxim tolerated dose [103,104,105,106]. Other strategies to reverse trastuzumab resistance via hyperactivation of the PIK/AKT/mTOR pathway have been attempted. One example is the BOLERO-1 phase III trial that evaluated the efficacy of adding an mTOR inhibitor, such as everolimus, to trastuzumab plus paclitaxel in the first-line treatment of HER2+ tumors. There was no significant difference in PFS between the two groups in the overall population. However, in the HR- group, there was a 7.2-month increase in PFS with everolimus [107].

Over the last five years, there have been breakthroughs in ER+ BC disease. Cyclines and CDKs are crucial for the regulation of cell cycle progression. CDKs are kinases that are regulated by their interactions with cyclines and CDK inhibitors. CDK activity is dysregulated in many breast cancer cells including HER2+, therefore, inhibiting CDKs might be a therapeutical option in this setting. Three different CDK 4/6 inhibitor drugs have been approved: Abemaciclib, Palbociclib and Ribociclib. They have all demonstrated increased PFS when combined with hormonotherapy in HER2-/ER+ [108,109,110]. Furthermore, Ribociclib, in combination with letrozole, achieved a 12-month increase in OS, reaching a median OS of 53.7 months for the ribociclib in the combination arm vs. 41.5 months in the letrozole plus placebo arm (HR, 0.73; 95% CI 0.59–0.90) [111]. Although CDK 4/6 suppresses Rb phosphorylation and blocks the cell cycle in the G1 phase, other mechanisms of action have been suggested. First, selective CDK4/6 inhibitors promote anti-tumor immunity by increasing type III interferon production and enhancing tumor antigen presentation. Indirect DNA methyltransferase-1 inhibition from CDK 4/6 inhibition also increases T cell cytotoxic activity and decreases immunosuppressive T reg cell [112]. Secondly, it also reduces TSC2 phosphorylation and attenuates mTORC1 activity, which could sensitize tumors to blocked EGFR/HER2. In PDX models, CDK4/6 inhibitors help sensitize HER-2+ breast tumors to HER2-targeted therapies [113]; however, there are no reliable biomarkers to predict the benefit of CDK 4/6 inhibition. The Basal intrinsic subtype based on PAM50 may be the only one showing no improvement with CDK 4/6 inhibitors in HER2-/ER+ patients. In a retrospective analysis of MONALEESA 2, 3 and 7 with 1,160 tumors, the basal-like subtype subgroup (n = 30) was the only one that did not benefit from the addition of Ribociclib (HR 1.15, *p* = 0.77) [114].

On the other hand, some HER2+/ER+ patients might benefit from dual pathway inhibition with the combination of HER2-targeted therapy plus CDK 4/6 inhibitors plus hormonotherapy. Some trials have already tested this triple combination in HER2+/ER+ patients. In the MonarchHER phase II trial, the triple combination of CDK inhibitor + hormonotherapy + antiHER2 (abemaciclib + fulvestrant + trastuzumab) increased PFS compared to other combinations. At the moment, some ongoing phase III trials, like PATINA, will help us understand which patients could benefit more from the triple combination and thus improve treatment personalization (Table 3).

## 4. Tumor Immunity and Immunotherapy in HER2-Positive Breast Cancer

### 4.1. Role of Tumor Immunity in HER2-Positive Breast Cancer

The role of immunotherapeutic agents in HER2+ BC is becoming increasingly relevant as this BC subgroup has higher stromal TIL levels, implying that HER2+ disease is usually more immunogenic compared with other BC tumors [115,116,117]. Thus, several immunotherapeutic agents, such as novel HER2-directed mAbs, ADCs, vaccines and adoptive T cell therapies are currently being explored in patients with HER2+ tumors. However, not all HER2+ tumors are equally immunogenic and specific BC molecular subgroups beyond immunohistochemical (IHC) subtypes show differential responses. Interestingly, HER-2 subgroup is more immunogenic than Luminal A/B [118]. Both the percentage of TILs and the expression of different immune cells in the tumor microenvironment (CD8+, CD4+ Th1 and NK cells) have been associated with better prognosis; they may also contribute to the therapeutic effects of anti-HER2 targeted therapy [119]. 

In HER2+ BC, the interaction between the immune system and the tumor is complex and dynamic, involving different HER2-targeted treatments with chemotherapy and hormonotherapy, which modulate the action of HR status and tumor biology [116]. In addition, the different anti-HER2 treatments seem to modulate the tumor microenvironment and vice versa and the presence of high tumor immunity has been linked to a differential effect of these therapies [72,120]. For example, in vivo models, lapatinib stimulates tumor infiltration by CD4 + CD8 + IFN-γ-producing T cells via a STAT1-dependent pathway. The deficiency of STAT1 decreases the therapeutic efficacy of lapatinib, remarking the importance of immune activation in the lapatinib antitumor activity [121]. Another example, from the Neo ALTTO trial, is that T cell-driven immune signatures have been associated with pCR in patients treated with lapatinib, which highlights the role of immunity in modulating the activity of HER2-targeted therapy [121,122]. The expression of PD1 in the tumor microenvironment is a mechanism of tumor evasion, the combination of anti-HER2 monoclonal antibodies with anti-PD1 is synergic, improving therapeutic activity, reason why these combinations are being tested in different trials [123]. 

Furthermore, the modulation of immune cells occurring in HER2+ BC has shown a clinical impact on treatment efficacy [124,125]. In HER2+ BC, the NeoSphere [126] and NeoALTTO [127] trials have shown that tumors with low baseline TILs had lower pCRrates. Additionally, both NeoALTTO [127] and TRYPHAENA [49] trials found that TILs were associated with improved event-free survival when systemic therapy was given in the neoadjuvant setting. Loi et al., reported that TILs were predictive of benefit to adjuvant trastuzumab in the FinHER study [128]. A pooled analysis of six prospective neoadjuvant clinical trials found that increased TIL levels were associated with higher pCR rates and improved DFS in HER2+ BC [129]; however, the analysis did not show an association between increased TIL levels and OS. In contrast, in the adjuvant N9831 trial, patients who received chemotherapy alone, the presence of high TIL levels was significantly associated with an improvement in recurrence-free survival; on the other hand, this benefit was not seen in patients treated with chemotherapy plus trastuzumab [130]. In summary, higher levels of TILs have been correlated with better outcomes and response to anti-HER2 treatment; however, an enhanced understanding of the role played by the immune system in modulating therapy response to different anti-HER2 agents is still needed.

Due to the promising role of immunotherapy in HER2+ BC, the recent introduction of immune checkpoint inhibitors and other immunotherapeutic agents capable of unleashing an anti-tumoral immune response opens new possibilities for therapeutic combinations in this setting.

### 4.2. Trials Testing Combinations with Immune-Therapeutic Drugs in HER2-Positive Breast Cancer

The most important trials testing immunotherapeutic agents in HER2+ disease are reported in Table 4.

Immune-therapeutic agents have mainly been tested and approved for triple-negative BC patients. Their use in other BC subgroups, such as luminal B BC or pretreated HER2+ BC, has also been explored but the results are still controversial [13,14,131,132]. The potential for immunotherapy to improve clinical outcomes in HER2 + disease have been explored in several preclinical and clinical studies.

CTLA-4 inhibitors have been little tested in BC. For example, a small trial, with nineteen patients, evaluated ipilimumab with or without cryoablation, in early-stage BC patients; two patients had HER2+ disease [123]. An increase of activated effector T cells was detected among patients receiving ipilimumab treatment alone or with cryoablation. Furthermore, the combination therapy also showed at increase in the ratio of intratumoral T effector cells relative to Tregs [131].

Controversial results were reported for anti-PDL1 treatment in HER2+ BC [13,132]. PANACEA, a single-arm, multicenter, phase Ib-2 trial, tested pembrolizumab plus trastuzumab in PD-L1-positive advanced HER2+ BC tumors. It was shown to be safe with durable activity and clinical benefit in PD-L1-positive, HER2+ trastuzumab resistant ABC patients [13].

KATE2, a randomized, double-blind, placebo-controlled, phase II study, included 202 treated patients, 133 with atezoizumab and 69 with placebo (n = 69), plus T-DM1 in HER2+ ABC. All patients had previously been treated with trastuzumab and a taxane. Atezolizumab plus T-DM1 did not generate a clinically substantial improvement in PFS and additionally was associated with extra adverse events [14].

In the JAVELIN Solid Tumor trial, avelumab was tested in 168 patients with metastatic BC, in which 26 patients (15.5%) had HER2+ BC. Unfortunately, no benefit in terms of overall response rate was seen in HER+ patients.

Based on these results, further studies in less pretreated HER2+ BC patients should focus on a PD-L1-positive population [133]. Another phase I trial, which only included 15 patients with HER2 metastatic BC, examined trastuzumab in combination with durvalumab. Disappointingly, none of these patients achieved a partial response and only four of them achieved stable disease; however, none had tumors harboring PD-L1 expression [134].

Further studies testing other immune-checkpoint inhibitors in combination with chemotherapy and anti-HER2 treatments are being explored to better characterize the efficacy of immune-checkpoint inhibitors in this setting (Table 4).

Beyond immune checkpoint inhibitors, HER2 is a remarkable therapeutic target for peptide-based cancer vaccines. Therapeutic cancer vaccines are intended to treat existing tumors by enhancing the anti-tumor immune response, and HER2 vaccination is currently under development using several different clinical development strategies. The use of dendritic cells (DCs) as a vaccine strategy has the advantage of presenting vaccine antigens to other immune system cell types. Furthermore, some preclinical studies have shown the possibility of generating HER2-loaded DCs as well as DCs engineered to express HER2 antigen epitopes [135,136].

Finally, the design and development of different HER2-directed ADCs explained above (Section 2) are also key immune-related strategies, which produce increased cytotoxicity with a reduction in chemotherapy off-target adverse effects due to the antibody-driven drug internalization.

## 5. Conclusions

In conclusion, the success of clinical research, as well as the better understanding of the biological heterogeneity of HER2+ disease over the years, has led to the introduction of several approaches in therapy for both the early and metastatic setting. These approaches include the exploration of new strategies towards a more personalized medicine, as are both escalated and deescalated strategies in the early setting; but also, the incorporation of new TKI and immune conjugates in the advanced setting changing the treatment sequence on this scenario. As future perspectives for HER2 positive breast cancer, we have focused on ER+/HER2+ BC disease and the role that immune response modulation plays in treatment. There is different evidence, both from a clinical and translational perspective, that suggests that HER2+/HR+ breast cancer should be considered as a different entity from HER2+/HR- disease. These data justify the design of specific trials for this population as it has been reviewed. On the other hand, the modulation of immune cells occurring in HER2+ BC has shown a clinical impact on treatment efficacy, and due to the promising role of immunotherapy in HER2+ BC, different immune checkpoint inhibitors and other immunotherapeutic agents are being tested in clinical trials opening new possibilities for therapeutic combinations in this setting. In summary, the higher level of understanding of the underlying molecular biology and the justification of the different available strategies will allow us to avoid overtreatment in some patients, relapses in other patients with early-stage disease, and finally, to improve treatment in patients with advanced disease. 

## Figures and Tables

**Figure 1 cancers-14-03305-f001:**
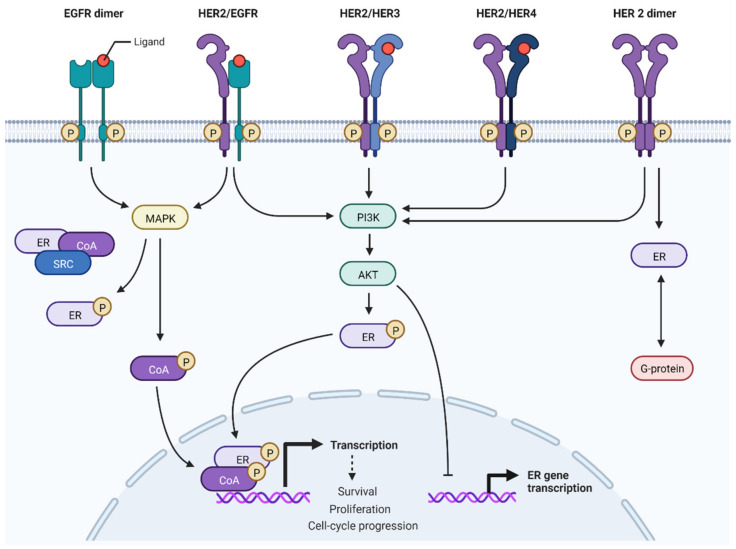
Interaction between estrogen receptor (ER) and human epidermal growth factor -2 (HER2) pathways. In ER+/HER2+ tumors, the hyperactivation of HER2 signaling activates downstream kinases and downregulates ER expression, not only at the protein level but also at mRNA expression. EGFR: epidermal growth factor receptor; HER3: human epidermal growth factor -3; HER4: human epidermal growth factor -4; P: phosphorylated; MAPK: mitogen-activated protein kinase; PI3K: phosphoinositide-3-kinase; CoA: coactivator; G-protein: Guanine nucleotide binding protein; AKT: protein kinase B; SRC: steroid receptor coactivator. (Adapted from “HER2 Signaling Pathway”, by BioRender.com (2022). Retrieved from https://app.biorender.com/biorender-templates, accessed on 1 July 2022).

**Table 1 cancers-14-03305-t001:** Trials in the neoadjuvant setting.

Trial	Phase	Treatment	n	pCR	DFS	OS
TRAIN-2	III	A: FEC + H + P → Pac + CBDA + H + P B: Pac + CBDA	438	A: 67 (60–73) B: 68 (61–74)	A: 3 y EFS: 92.7 (89.3–96.2) B: 3 y EFS: 93.6 (90.4–96.9)	A: 3 y: 97.7 (95.7–99.7) B: 3 y: 98.2 (96.4–100)
WSG-ADAPT	II	A: (H + P) B: (H + P) + Pac	134	A: 34.4 (24.7–45.2) B: 90.5 (77.4–97.3)		
KRISTINE	III	A: T-DM1 + P B: TCH + P	444	A: 44 B: 56	A: 3 y IDFS: 93 (89.4–96.7) B: 3 y IDFS: 92 (86.7–97.3)	A: 3 y: 97 (94.6–99.4) B: 3 y: 97.6 (95.5–99.7)
TBCRC006	II	Lapatinib + H	64	27		
TBCRC023	II	A: Lapatinib + H B: Lapatinib + H	97	A: 12 B: 28		
PER-ELISA	II	A: Letrozole + H + P B: Pac + H + P	61	A: 20.5 (11.1–34.5) B: 81 (57–93.4)		
SOLTI-PAMELA	II	Lapatinib + H	151	30 (23–39)		
NeoSphere	II	H + P	107	16.8 (10.3–25.3)	5 y DFS: 80 (70–86)	
PHERGain	II	A: TCHP B: HP	356	A: 57.7 (47.4–69.4) B: 35.4 (29.9–41.3)		
NeoSphere	II	A: H + T B: P + H + T C: P + H D: P + T	417	A: 29 (20.6–38.5) B: 45.8 (36.1–55.7) * C: 16.8 (10.3–25.3) † D: 24.0 (15.8–33.7) ‡ * *p* = 0.0141 vs. A † *p* = 0.0198 vs. A ‡ *p* = 0.003 vs. B	5 y DFS A: 81 (71–87) B: 86 (77–91) C: 73 (64–81) D: 73 (63–81)	
TRYPHAENA	II	A: FEC + H + P→ THP B: FEC→ THP C: TCHP	225	A: 56.2 B: 54.7 C: 63.6		
BERENICE	II	A: ddAC→Pac + H + P B: FEC → THP	400	A: 61.8 (54.7–68.6) B: 60.7 (53.6–67.5)		
Meta-analysis (CALGB 40601, Cher-LOB, NSABP-B41, NeoALTTO)	II/III	A: ChT + H B: ChT + H + Lapatinib	1410		RFS Pooled HR 0.62 (0.46–0.85)	OS Pooled HR 0.65 (0.43–0.98)
Cher-LOB	II	A: ChT + H B: ChT + Lapatinib C: ChT + H + Lapatinib ChT: Pac → FEC	121	A: 25 (13.1–36.9) B: 26.3 (14.5–38.1) C: 46.7 (34.4–58.9)	A: 5 y RFS: 77.8 B: 5 y RFS: 77.1 C: 5 y RFS: 85.5 HR 0.52 (0.23–1.15) (A vs. C)	HR 1.00 (0.31–3.27) (A vs. C)
NSABP-B41	III	A: ChT + H B: ChT + Lapatinib C: ChT + H + Lapatinib ChT: AC → Pac	529	A: 52.5 (44.9–59.5) B: 53.2 (45.4–60.3) C: 62 (54.3–68.8)	A: 5 y RFI: 84.3 B: 5 y RFI: 78.6 C: 5 y RFI: 90 EFS: HR 0.66 (0.34–1.25) (A vs. C)	A: 5 y: 94.5 B: 5 y: 89.4 C: 5 y: 95.7 HR 1.00 (0.24–1.67) (A vs. C)
NeoALTTO	III	A: H + ChT B: Lapatinib + ChT C: H + Lapatinib + ChT ChT: Pac	455	29.5 (22.4–37.5) 24.7 (18.1–32.3) 51.3 (43.1–59.5)	6 y EFS: 67 6 y EFS: 67 6 y EFS: 74 EFS: HR 0.98 (0.64–1.91) (A vs. C)	6 y: 82 6 y: 79 6 y: 85 HR 0.85 (0.49–1.86) (A vs. C)
CALGB 40601	III	A: H + ChT B: Lapatinib + ChT C: H + Lapatinib + ChT ChT: Pac	305	46 (37–55) 32 (22–45) 56 (47–65)	7 y EFS: 79 7 y EFS: 69 7 y EFS: 93 EFS: HR 0.32 (0.14–0.71) (A vs. C)	7 y: 88 7 y: 84 7 y: 96 HR 0.34 (0.12–0.94) (A vs. C)

T-DM1: trastuzumab-emtansine; ChT: chemotherapy; Pac: paclitaxel; H: trastuzumab; P: Pertuzumab: T: docetaxel; A: adriamycin; C: cyclophosphamide; dd: dose dense; CBDA: carboplatin; F: 5-fluorouracil; E: epirubicin; pCR: pathologic complete response; DFS: disease-free survival; OS: overall survival; EFS: events free survival; RFI: relapse free interval; RFS: relapse free survival; IDFS invasive disease-free survival; y: year. pCR, DFS and OS are % (95% CI).

**Table 2 cancers-14-03305-t002:** Trials in the adjuvant setting.

Trial	Phase	Treatment	n	DFS	OS
HERA	III	A: 1 y H B: 2 y H C: Observation	3389	A: 2 y DFS: 85.8 C: 2 y DFS: 77.4 *p* < 0.0001 A: 10 y DFS: 69 B: 10 y DFS: 69 C: 10y DFS: 63 HR 0.76 (0.68–0.86) (A vs. C)	A: 2 y: 96 C: 2 y: 95.1 *p* = 0.26 A: 12 y: 79 B: 12 y: 80 C: 12 y: 73 HR 0.74 (0.64–0.86) (A vs. C)
BCIRG-006	III	A: AC →T B: AC → T → 1 y H C: TC → 1 y H	3222	A: 5 y DFS: 75 B: 5 y DFS: 84 C: 5 y DFS: 81 *p* < 0.001 (A vs. B) *p* = 0.04 (A vs. C) A: 10 y DFS: 67.9 B: 10 y DFS: 74.6 C: 10 y DFS: 73 *p* < 0.0001 (A vs. B) *p* = 0.0011 (A vs. C)	A: 5 y: 87 B: 5 y: 92 C: 5 y: 91 *p* < 0.001 (A vs. B) *p* = 0.04 (A vs. C) A: 10 y: 78.7 B: 10 y: 85.9 C: 10 y: 83.3 *p* < 0.0001 (A vs. B) *p* = 0.0075 (A vs. C)
NSABP B-31	III	A: AC →Pac B: AC → Pac + 1 y H	3351	A: 10 y DFS: 62.2 B/C: 10 y DFS: 73.7 *p* ≤ 0.001 HR 0.6 (0.53–0.68)	A: 10 y OS: 75.2 B/C: 10 y OS: 85 *p* = 0.001 HR 0.63 (0.54–0.73)
NCCTG N9831	III	A: AC → Pac B: AC → Pac + 1 y H C: AC → Pac → 1 y H			
Short-HER	III	A: AC or EC → T or Pac → 1 y H B: T → FEC → 9 w H	1254	A: 5 year-DFS: 88.5 B: 5 year-DFS: 85.5HR 1.13 (90% CI, 0.89–1.42) 8.7 y DFS: HR 1.09 (90% CI, 0.88–1.35)	A: 5 y: 95.2 B: 5 y: 95 HR 1.07 (CI 90%, 0.74–1.56) A: 9 y: 90 B: 9 y: 91 HR 1.18 (CI 90%, 0.86–1.62)
SOLD	III	A: T + 9 w H →FEC B: T → FEC → 1 y H	2174	A: 5 y DFS: 88 B: 5 y DFS: 90.5 HR 1.39 (CI 90%, 1.12–1.72)	A: 5 y: 94.7 B: 5 y: 95.9 HR 1.36 (CI 90%, 0.98–1.89)
PHARE	III	A: 6 m H B: 12 m H	3384	A: 7.5 y DFS: 78.8 B: 7.5 y DFS: 79.6 *p* = 0.39 HR 1.08 (0.93–1.25)	7.5 y OS: HR 1.13 (0.92–1.39)
PERSEPHONE	III	A: ChT + 6 m H B: ChT + 1 y H	4089	A: 4 y DFS: 89.4 B: 4 y DFS: 89.8 *p* = 0.011 HR 1.07 (CI 90%, 0.93–1.24)	A: 4 y: 93.8 B: 4 y: 94.8 HR 1.14 (CI 90%, 0.95–1.37)
APT	II	Pac + 12 w H → 9 m H	406	7 y DFS: 93.3 (90.4–96.2)	7 y: 95 (92.4–97.7)
APHINITY	III	A: ChT + 1 y H B: ChT + H + 1 y P	4800	A: 6 y DFS: 88 B: 6 y DFS: 91 HR 0.76 (0.64–0.91)	A: 6 y: 94 B: 6 y: 95 *p* = 0.17 *Immature data*
ALTTO	III	A: ChT + H → lapatinib B: ChT + H + lapatinib C: ChT + H	8381	A: 6.9 y DFS: 84 B: 6.9 y DFS: 85 C: 6.9 y DFS: 82 HR 0.86 (0.74–1.0) (B vs. C) HR 0.93 (0.81–1.08) (A vs. C)	A: 6 y: 92 B: 6 y: 93 C: 6 y: 91 HR 0.86 (0.7–1.06) (B vs. C) HR 0.88 (0.71–1.08) (A vs. C)
KAITLIN	III	A: AC →T-DM1 + P B: AC → P + H + P	1846	A: 3 y DFS: 93 B: 3 y DFS: 94 *p* = 0.827 HR 0.98 (0.72–1.32)	*Immature data for OS*
ExteNET	III	A: Neratinib 1 y after H-based therapy B: Observation after H-based therapy	2840	5 y DFS in HR+/≤ 1 y post-H: A: 90.8 B: 85.7 HR 0.58 (0.41–0.82) 5 y DFS in HR+/>1 y post-H and residual disease after NA: benefit of 7.4% in group A vs. B HR 0.60 (0.33–1.07)	8y OS in HR+/≤ 1 y post-H: A: 91.5 B: 89.4 HR 0.79 (0.55–1.13) 8y OS in HR+/>1 y of prior H and residual disease after NA: benefit of 9.1% in group A vs. B HR 0.47 (0.23–0.92)
KATHERINE	III	A: T-DM1 × 14 cycles B: H × 14 cycles	1486	A: 3 y DFS: 88.3 B: 3 y DFS: 77 HR 0.5 (0.39–0.64)	*Immature data for OS*

T-DM1: trastuzumab-emtansine; T: docetaxel; Pac: paclitaxel; H: trastuzumab; P: pertuzumab; F: 5-fluorouracil; E: epirubicin; A: adriamycin; C: cyclophosphamide; ChT: chemotherapy; pCR: pathologic complete response; DFS: disease-free survival; OS: overall survival; y: year; m: month; w: week; NA: neoadjuvant treatment; HR+: hormone receptor positive pCR, DFS and OS are % (95% CI).

**Table 3 cancers-14-03305-t003:** Trials testing CDK4/6 inhibitors.

Trial	Phase	Treatment	*n*	Primary Endpoint	Results	Target
PATRICIA	II	Palbociclib 200 mg 2 w on 1 w off or 125 3 w on 1 w off + H 600 mg sc every 3 w ER+ patients were treated with Letrozole vs. Placebo	71 (56 ER+)	PFS at 6 m	6 m PFS in ER+ patients treated with Palbociclib + H 42.8% vs. 46.4% Luminal disease by PAM50 had longer PFS (10.6 m vs. 4.2 m)	Similar potency against CDK 4 than CDK 6
Ribociclib, NCT02657343	Ib/II	Ribociclib 400 mg daily (phase II) + H iv	13 (ER + in 8)	MTD and CBR	1 experienced stable disease >24 w PFS was 1.3 m	Greater potency against CDK 4 than CDK 6
Ribociclib, NCT02657343	Ib	Ribociclib 400 mg given on days 8–21 of a 21-day cycle with T-DM1	12	MTD for phase II	PFS was 10.4 m	Greater potency against CDK 4 than CDK 6
MonarchHER	II	A: Abemaciclib 150 mg/12 h + H iv B: Abemaciclib + BPC ChT C: Abemaciclib + fulvestrant im + H iv	237 physician’s choice (all HER+/ER+)	PFS between groups	Abemaciclib + H + Fulvestrant longer PFS: 8.3 m vs. 5.7 m and 5.7 m compared with the other groups	Greater potency against CDK 4 than CDK 6, also CDK 1/2/5 inhibitor
PATINA	III	H + P with endocrine therapy (letrozole, anastrozole, exemestane or fulvestrant) +/- palbociclib	496 already recruited	PFS	Not reported yet	Similar potency against CDK 4 than CDK 6
ASPIRE	I/II	Palbociclib (100 and 125 mg 3 w on 1 w off) + H iv + P iv + Anastrozole	36 planned	DLT, MTD, CBR	Not reported yet	Similar potency against CDK 4 than CDK 6

MTD: Maximum Tolerated Dose; PFS: Progression-Free survival; CBR: Clinical Benefit Rate; BPC: best physician’s choice; ROR: rate of overall response; DLT: dose limiting toxicity; iv: intravenous; im: intramuscular; sc: subcutaneous, H: trastuzumab; P: pertuzumab; ChT: chemotherapy; m: months; w: weeks.

**Table 4 cancers-14-03305-t004:** Trials testing immunotherapeutic agents in HER2+ disease.

NCI Identifier	Phase	Recruitment	Setting	Subtype	Immunotherapies	Combined Treatments	Target
NCT03241173	I/II	Active, not recruiting	Metastatic or LA	All	Ipilimumab and/or nivolumab	INCAGN01949	Anti-CTLA4
NCT03126110	I/II	Active, not recruiting	Metastatic or LA	All	Ipilimumab and/or nivolumab	INCAGN01876	Anti-CTLA5
NCT03328026	I/II	Recruiting	Metastatic or LA	All	Ipilimumab or pembrolizumab	SV-BR-1-GM, cyclophosphamide, and interferon inoculation	Anti-CTLA6
NCT02129556	I/II	Active, not recruiting	Metastatic	HER2+ resistant to H	Pembrolizumab	H	anti-PDL1; anti-PD1
NCT01772004	I	Active, not recruiting	Metastatic	HER2+	Avelumab	H	Anti-PD1
NCT03747120	II	Recruiting	Neoadjuvant	HER2+	Pembrolizumab	Neoadjuvant H + P + Pac	Anti-PDL1
NCT03523572	I	Recruiting	Advanced	HER2+	Nivolumab	Trastuzumab deruxtecan	Anti-PDL1; Anti-PD3
NCT02649686	I	Active, not recruiting	Metastatic	HER2+	Durvalumab	H	Anti-PDL1; Anti-PD4
NCT02924883	II	Active, not recruiting	Metastatic	HER2+	Atezolizumab	T-DM1	Anti-PDL1; Anti-PD5
NCT03125928	II	Recruiting	Metastatic	HER2+	Atezolizumab	Pac + H + P	Anti-PDL1; Anti-PD6
NCT03620201	I	Recruiting	Stage II–III	HER2+	M7824 (anti-PD-L1 fusion protein)		Anti-PDL1; Anti-PD7
NCT05180006	I	Recruiting	Neoadjuvant	HER2+ TNBC	Atezolizumab	H + P	Anti-PDL1
NCT02336984	I/II	Active, not recruiting	DCIS	HER2+	HER2-pulsed DC1	H + P	Vaccine
NCT02061423	I	Active, not recruiting	Stage I–III	HER2+	HER2-pulsed DC vaccine		Vaccine
NCT03384914	II	Recruiting	Stage I–III	HER2+	DC1 vaccine		Vaccine
NCT03387553	I	Active, not recruiting	Stage II/III	HER2+	DC1 vaccine		Vaccine
NCT03113019	I	Active, not recruiting	Stage II–IV	HER2+ TNBC	DC-based vaccine		Vaccine
NCT03113019	I	Active, not recruiting	Stage II–IV	HER2+ TNBC	DC-based vaccine		Vaccine
NCT03630809	II	Not yet recruiting	DCIS or inflammatory	HER2+	HER2-pulsed DC1		Vaccine
NCT01376505	I	Recruiting	Metastatic	HER2 1+, 2+, or 3+ by IHC	MVF-HER-2 (597–626)-MVF- HER-2 (266–296) peptide vaccine		Vaccine
NCT03632941	II	Recruiting	Metastatic	HER+	VRP-HER2 immunizations plus pembrolizumab.		Vaccine

T-DM1: trastuzumab-emtansine; Pac: paclitaxel; H: trastuzumab; P: pertuzumab; LA: locally advanced; DC: dendritic cells; IHC: immunohistochemistry; DCIS: ductal carcinoma in situ; TNBC: triple-negative breast cancer.

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
