# Peer review of "Targeted Therapeutic Options and Future Perspectives for HER2-Positive Breast Cancer"

_cancers, 2022, doi:10.3390/cancers14143305_

Round 1

Reviewer 1 Report

In this review the authors provide ample amount of information on current and new various targeted therapeutic options available/explored in HER2+ breast cancer and also tried to address the future perspectives in this area. They highlighted past therapies explored in recent years that has improved the prognosis of HER2+ early breast cancer. However, around 25% of EBC patients will still relapse. To overcome the HER+ treatment resistance different new combination therapies have been recently explored in clinical trials targeting MAPK and PI3K pathways, and CDK4/6 kinases downstream of HER2. Although these approaches are encouraging but it still needs conclusive outcomes. In addition to this the importance of crosstalk between ER and HER2 pathways in understanding the resistance mechanisms HER2+/ER+ BC is required for optimization of future drug combinations. The review also justifies the need for investigating the use of immunotherapeutic agents in HER2+ BC in some extreme cases.

The review was written well, easy to follow and justifies the significant advances in new targeted therapeutic approaches in HER2+ breast cancer. However, some minor improvements are needed:

11. Some concepts are repeated more than once, rather than being addressed the first time they are mentioned at several places such as targeting CDK4/6 kinases as a new approach to overcome resistance in HER+ BC.

22. Few sentences are repetitive in many places in the manuscript. As an example, lines 36 & 71; line 33-35 & 68-71. Try to rephrase the sentence or elaborate in a different way so that it looks different from the abstract.

 3. Wherever it is necessary I suggest authors should also explain in brief about proposed molecular mechanisms (crosstalk with other receptor signaling pathways) reported in the literature for resistance to HER2 drugs/relapse in HER+ breast cancer patients and how these are being addressed in clinical trials. For instance, they can mention recent developments indicating the importance of cross talk between PRLR and HER2 signaling as an alternative route that can further promote tumor growth and may be responsible for resistance in HER2+ BC resulting in tumor malignancy. For this a combination therapy targeting PRLR (G129R) and HER2+ indicated by Scotti et al., 2008. And other studies like Notch-EGFR/HER2 bidirectional crosstalk in breast cancer and functional crosstalk between HER2 and TGF-β signaling in breast cancer malignancy was also reported in the past. Although there no ongoing trials targeting these receptors along with HER2 but these studies may provide better screening for the expression of these oncogenes and optimize their personal treatment options and predict potential treatment resistance.

e 4. The future perspectives for HER2+ BC seems to be spread out everywhere in the text and not clearly highlighted/visible. They may put all together in a separate paragraph under the heading "future perspectives for HER2+ BC".

4 5.  Line 41: ‘PDX model’ expand PDX as patient derived xenograft mice model.

5 6. Line 279: Indicate triple positive tumors means (ER+PR+ & HER+).

Author Response

In this review the authors provide ample amount of information on current and new various targeted therapeutic options available/explored in HER2+ breast cancer and also tried to address the future perspectives in this area. They highlighted past therapies explored in recent years that has improved the prognosis of HER2+ early breast cancer. However, around 25% of EBC patients will still relapse. To overcome the HER+ treatment resistance different new combination therapies have been recently explored in clinical trials targeting MAPK and PI3K pathways, and CDK4/6 kinases downstream of HER2. Although these approaches are encouraging but it still needs conclusive outcomes. In addition to this the importance of crosstalk between ER and HER2 pathways in understanding the resistance mechanisms HER2+/ER+ BC is required for optimization of future drug combinations. The review also justifies the need for investigating the use of immunotherapeutic agents in HER2+ BC in some extreme cases.

The review was written well, easy to follow and justifies the significant advances in new targeted therapeutic approaches in HER2+ breast cancer. However, some minor improvements are needed:

  1. Some concepts are repeated more than once, rather than being addressed the first time they are mentioned at several places such as targeting CDK4/6 kinases as a new approach to overcome resistance in HER+ BC.

RESPONSE: Thank you for your response. As per your suggestion, we have now addressed this point by removing several repeated sentences over the manuscripts such as those in lines: 97, 136, 231, 279, 284, 312, 315, 336, 449 and 604.

  1. Few sentences are repetitive in many places in the manuscript. As an example, lines 36 & 71; line 33-35 & 68-71. Try to rephrase the sentence or elaborate in a different way so that it looks different from the abstract.

RESPONSE: Thank you for your time. We have modified the text accordingly.

  1. Wherever it is necessary I suggest authors should also explain in brief about proposed molecular mechanisms (crosstalk with other receptor signaling pathways) reported in the literature for resistance to HER2 drugs/relapse in HER+ breast cancer patients and how these are being addressed in clinical trials. For instance, they can mention recent developments indicating the importance of cross talk between PRLR and HER2 signaling as an alternative route that can further promote tumor growth and may be responsible for resistance in HER2+ BC resulting in tumor malignancy. For this a combination therapy targeting PRLR (G129R) and HER2+ indicated by Scotti et al., 2008. And other studies like Notch-EGFR/HER2 bidirectional crosstalk in breast cancer and functional crosstalk between HER2 and TGF-β signaling in breast cancer malignancy was also reported in the past. Although there no ongoing trials targeting these receptors along with HER2 but these studies may provide better screening for the expression of these oncogenes and optimize their personal treatment options and predict potential treatment resistance.

RESPONSE: Thank you for your great suggestion. We have now expanded the explanations where needed and included explanation on the different molecular mechanisms in regards of these studies as per your suggestion in lines 134 and 370.

  1. The future perspectives for HER2+ BC seems to be spread out everywhere in the text and not clearly highlighted/visible. They may put all together in a separate paragraph under the heading "future perspectives for HER2+ BC".

RESPONSE: Thank you for your acknowledgement on this. Our distribution is based on the different drug mechanisms rather than current vs new approached. Thus, we include both current and future perspectives in each of the sections. However, to address your concern, we have tried to highlight better the parts of the text in which we talk about future perspectives.

  1. Line 41: ‘PDX model’ expand PDX as patient derived xenograft mice model.

       RESPONSE: We have now addressed this typo.

  1. Line 279: Indicate triple positive tumors means (ER+PR+ & HER+).

RESPONSE: We have now addressed this in the text.

Reviewer 2 Report

The manuscript by Ferrando-Díez et al presents a review focusing on targeted therapeutic options and future perspectives for HER2 positive breast cancer. Therapeutics targeting Her2+ breast cancer have significantly improved the clinical outcomes of patients with this subtype of breast cancer. However, therapeutic resistance and cancer relapses remain a significant challenge.  In the review, the authors reviewed various therapeutic options/regimens for HER2+ breast cancer based on different settings, which is  highly clinically  relevant.  The manuscript includes  extensive literature citation/summary, and have a comprehensive presentation of pertinent issues. However, it would be strengthened if the following questions could be addressed.

1. The review mentioned  numerous therapeutic agents. In contrast to clear presentation of the drugs/agents in the early parts,  the drugs/agents in the second half of the paper were compounded with numerous trial names,  and were less clear. It would be better to have a summary table of individual  therapeutics in groups based on different mechanisms.   

2. It may be necessary to go a little more details of drug action site for Her2 targeting agents. For example, clarifying that pertuzumab targets HER3 may help to understand the "double blocking" strategies and the resulted benefits.

3. It would be better to have a concise introduction of HER2-mediated pathogenesis and targeting strategy before talking about current approaches. The Figure 1 could be modified and moved here for this purpose. The figure could also be cited later in the mechanistic discussion.

4. Figure 1 has several accuracy issues. e.g. 1) do they want to say  "ER-G protein" interaction downstream of membrane associated activation of ER?; 2) "ER gene transcription" inhibited by Akt in cytosolic space  is confusing.

5. The perspectives may include both novel trials/regimens and novel Her2 targeting agents that are being developed/tested/

6. The conclusion part could be more informative and solidified.

Author Response

The manuscript by Ferrando-Díez et al presents a review focusing on targeted therapeutic options and future perspectives for HER2 positive breast cancer. Therapeutics targeting Her2+ breast cancer have significantly improved the clinical outcomes of patients with this subtype of breast cancer. However, therapeutic resistance and cancer relapses remain a significant challenge.  In the review, the authors reviewed various therapeutic options/regimens for HER2+ breast cancer based on different settings, which is  highly clinically  relevant.  The manuscript includes  extensive literature citation/summary, and have a comprehensive presentation of pertinent issues. However, it would be strengthened if the following questions could be addressed.

  1. The review mentioned  numerous therapeutic agents. In contrast to clear presentation of the drugs/agents in the early parts,  the drugs/agents in the second half of the paper were compounded with numerous trial names,  and were less clear. It would be better to have a summary table of individual  therapeutics in groups based on different mechanisms.   

RESPONSE: Thank you for your time and your great suggestion. As you mentioned, in the first part of the article there is a presentation of the agents directed against HER2. However, the mechanism of action of the drugs in the second part (CDK inhibitors and immunotherapy) are not explained in the text. Regarding immunotherapy, the mechanisms of action are included in Table 4 (last column). In contrast, in the previous version, there was not any explanation about the mechanism of action of CDK inhibitors, so we have added a column in Table 3 and a brief presentation in the text (lines 467-473).

  1. It may be necessary to go a little more details of drug action site for Her2 targeting agents. For example, clarifying that pertuzumab targets HER3 may help to understand the "double blocking" strategies and the resulted benefits.

RESPONSE: Thank you for the acknowledgement. We have added this explanation in line 106.

  1. It would be better to have a concise introduction of HER2-mediated pathogenesis and targeting strategy before talking about current approaches. The Figure 1 could be modified and moved here for this purpose. The figure could also be cited later in the mechanistic discussion.

RESPONSE: Thank you for your suggestion. We have expanded on the pathogenesis of HER2 BC at the beginning of part 2, line 97.

  1. Figure 1 has several accuracy issues. e.g. 1) do they want to say  "ER-G protein" interaction downstream of membrane associated activation of ER?; 2) "ER gene transcription" inhibited by Akt in cytosolic space  is confusing.

RESPONSE: Thank you for your time. 1) yes, we want to say the interaction between ER and G protein downstream of membrane associated activation of ER; 2) we have modified Figure 1 in order to make this part clearer.

  1. The perspectives may include both novel trials/regimens and novel Her2 targeting agents that are being developed/tested/

RESPONSE: Thank you for your acknowledgement on this. Novel trials with immunotherapy and CDK inhibitors are addressed in sections 3.3 and 4.2. Regarding novel HER2 targeting agents that are being developed, we mention some of them in section 2.2, such as margetuximab and trastuzumab- duocarmazine. Do not hesitate to suggest as any further information you think might be missing on the manuscript.

  1. The conclusion part could be more informative and solidified.

RESPONSE: Thank you for your suggestion. We have rewritten the conclusion according to your recommendation.